**Data Availability Statement:** All relevant data are within the manuscript and its Supporting Information files.

**Funding:** This study was funded by a seed grant from the Environmental Center of the University of

# A systematic review of waterborne and water-related disease in animal populations of Florida from 1999–2019

**Meg Jenkins**[ORCID], **Sabrina Ahmed**[ORCID], **Amber N. Barnes**[ORCID]*

Department of Public Health, University of North Florida, Jacksonville, Florida, United States of America

* amber.barnes@unf.edu

## Abstract

### Background

Florida's waters are a reservoir for a host of pathogens and toxins. Many of these microorganisms cause water-related diseases in people that are reportable to the Florida Department of Health. Our objective in this review was to ascertain which water-related pathogens and toxins of public health importance have been found in animal populations in Florida over the last twenty years.

### Methods

Nineteen databases were searched, including PubMed and Web of Science Core Collection, using keywords and search terms for the waterborne diseases, water-related vector-borne diseases, and water-based toxins reportable to the Florida Department of Health. For inclusion, peer-reviewed journal articles were to be written in English, published between January 1, 1999 and December 31, 2019, and contain primary research findings documenting at least one of the water-related pathogens or toxins of interest in an animal population within Florida during this same time frame.

### Results

Of over eight thousand initial search results, 65 studies were included for final analysis. The most common animal types implicated in the diseases of interest included marine mammals, fish and shellfish, wild birds, and livestock. Toxins or pathogens most often associated with these animals included toxin-producer *Karenia brevis*, vibriosis, *Escherichia coli*, and Salmonellosis.

### Discussion/conclusion

Findings from this review elucidate the water-related disease-causing pathogens and toxins which have been reported within animal populations in recent Florida history. As most of these diseases are zoonotic, our results suggest a One Health approach is necessary to support and maintain healthy water systems throughout the state of Florida for the protection of both human and animal populations.

North Florida to ANB. https://www.unf.edu/ecenter/
The funders had no role in study design, data
collection and analysis, decision to publish, or
preparation of the manuscript.

**Competing interests:** The authors have declared
that no competing interests exist.

## Introduction

The state of Florida is a peninsula with a unique environment consisting of hundreds of miles of coastline on the Atlantic Ocean and Gulf of Mexico, along with many inland freshwater bodies. Florida's waters are an important source of economic revenue for the state, in both recreational and commercial sectors. In 2016 these combined industries brought in $27.8 billion dollars and supported over 170,000 jobs [1]. Freshwater-based recreational tourism from just the St. Johns River basin alone is estimated to bring in over $200 million dollars a year [2]. Florida's beaches and lakes are hotspots for water activities such as fresh and saltwater fishing, boating, surfing, diving and kayaking for both local residents and out-of-state visitors alike. Each year, the state boasts over 130 million visitors [3].

Florida is home to over 20 million people and numerous species of novel, native and imperiled wildlife [4, 5]. Some of these animals include the American alligator, the American crocodile, alligator snapping turtle, sea turtles (i.e. leatherback, Hawksbill, Kemp's Ridley, green sea turtle, and loggerhead), whales (i.e. North Atlantic Right, Finback, Sei, sperm, and humpback), many types of birds such as brown pelicans and burrowing owls, various fresh and saltwater fish, amphibians, and coral (i.e. staghorn coral), snakes and other land reptiles (i.e. gopher tortoise), land mammals (i.e. Everglades mink, red wolf, Florida bonneted bat, key deer, and Florida panther), and marine mammals (i.e. Florida manatee) [5]. Preserving and protecting the biodiversity of Florida is critical to maintaining the health of the environment, animals and residents and vital to the recreation and hospitality sectors [6, 7].

The environmental conditions of Florida and its many waterways create favorable ecosystems and transmission opportunities for various water-related infections that affect both humans and animals. Waterborne infectious diseases are primarily spread through the ingestion of contaminated water, often due to poor human and animal waste management, and can cause negative symptoms in both humans and animals if ingested [8]. Water-related vector-borne diseases are also caused by pathogenic microbes but transmission to humans or animals requires a vector that uses water in its life cycle, such as a mosquito. Finally, water-related toxins are harmful substances which are naturally produced by organisms that reside in the water and have the ability to contaminate water bodies [8]. Water-related pathogens are often zoonotic, meaning they can be transmitted between an animal and a person, and create significant public health challenges for prevention, detection and control [9].

Because some zoonotic water-related pathogens and toxins can cause serious, sometimes fatal, human disease and have the potential for person-to-person transmission, cases of human infections are reportable to the Florida Department of Health by the documenting physician [10]. In addition, several of these same diseases discovered in animal populations are also reportable to the Florida Department of Agriculture and Consumer Services by attending veterinarians [11]. Yet these reportable water-related diseases vary widely in their exposure risks, their transmission routes, and their disease presentations. These factors make it difficult to coordinate surveillance efforts between state public health and veterinary professionals or develop a universally-applicable approach to decrease their burden. The objective of this study is to identify which waterborne diseases, water-based toxins, and water-related vector-borne diseases of public health importance have been reported in animal species of Florida over the last twenty years. Although the included studies do not correlate directly to human disease risk or prevalence, understanding which water-related pathogens and toxins impact Florida's animals in recent years can illuminate ways to prevent transmission in both populations.

## Methods

Florida statute (Rule 64D-3.029. Florida Administrative Code) mandates that over 60 human diseases are reportable to the state Department of Health [10, 12]. Of these, 33 are infectious

**Table 1. Select reportable water-related diseases in Florida analyzed in this study.**

| Waterborne | Water-Related | Water-Based Toxins |
|---|---|---|
| | Vector-Borne | |
| • Amebic encephalitis<br>• Campylobacteriosis<br>• Cholera (Vibrio cholerae type O1)<br>• Cryptosporidiosis<br>• Cyclosporiasis<br>• Escherichia coli infection, Shiga toxin producing<br>• Giardiasis<br>• Hepatitis A<br>• Hepatitis E<br>• Legionellosis<br>• Leptospirosis<br>• Melioidosis<br>• Poliomyelitis*<br>• Salmonellosis<br>• Shigellosis<br>• Tularemia<br>• Typhoid fever (Salmonella serotype Typhi)<br>• Vibriosis (Vibrio species and closely related organisms, not Vibrio cholerae type O1) | • Arboviral diseases not otherwise listed<br>• California serogroup virus disease<br>• Chikungunya fever<br>• Dengue fever<br>• Eastern equine encephalitis<br>• Malaria<br>• St. Louis encephalitis<br>• Venezuelan equine encephalitis<br>• Viral hemorrhagic fevers<br>• West Nile virus disease<br>• Yellow fever<br>• Zika fever | • Ciguatera fish poisoning<br>• Neurotoxic shellfish poisoning<br>• Saxitoxin poisoning (paralytic shellfish poisoning) |

*Humans are the only known reservoir

pathogens which can be identified as either a waterborne disease, water-related vector-borne disease, or water-based toxin. Based on these designations, these diseases were identified to be of public health importance, and are investigated here to ascertain whether they are found in animal populations in Florida. The diseases of interest for this study have been sorted according to their categorial classification (Table 1).

## Search strategy and inclusion criteria

Predefined PRISMA guidelines [13] were used to identify, screen, and assess publications for eligibility. The PRISMA Statement provides a flow diagram (Fig 1) and checklist (S1 File) which can be used to map the systematic process of screening titles for inclusion in either a systematic review or meta-analysis [13]. Nineteen online databases were searched. These included PubMed, Web of Science Core Collection, Google Scholar, ProQuest databases of ABI/ INFORM, Agriculture Collection, Agricola, Aquatic Science Collection, Biological Sciences, Agricultural and Environmental Science Collection, Health & Medicine, MEDLINE, and TOXLINE, GALE databases of Nursing and Allied Health Outcomes and Environmental Studies and Policy, SAGE Journals, Science Direct, SpringerLink Journals, BioOne Complete, and Environment Complete. Search strings were developed for each database containing keywords related to the waterborne, water-based toxins, and water-related vector-borne diseases reportable to the Florida Department of Health [FDOH; 10] using Boolean Operators and wildcard symbols(*) as appropriate for each database. An example of a search string is included below:

a. ("waterborne disease" OR "water-based vector" OR "water-related toxin" OR "Amebic encephalitis" OR "Naegleria fowleri" OR "Primary amebic meningoencephalitis" OR arbovir* OR "California serogroup" OR "Jamestown Canyon" OR "Snowshoe Hare Virus" OR "La Crosse Virus" OR Campylobacter* OR Chickungunya OR Cholera OR "Vibrio cholerae" OR Ciguatera OR Cryptosporidi* OR Cyclospor* OR Dengue OR "Eastern Equine Encephalitis" OR "Escherichia coli" OR "Shiga Toxin" OR STEC OR Giardi* OR "Hepatitis A" OR "Hepatitis E" OR Legionell* OR Leptospir* OR Weil OR Malaria* OR Melioidosis

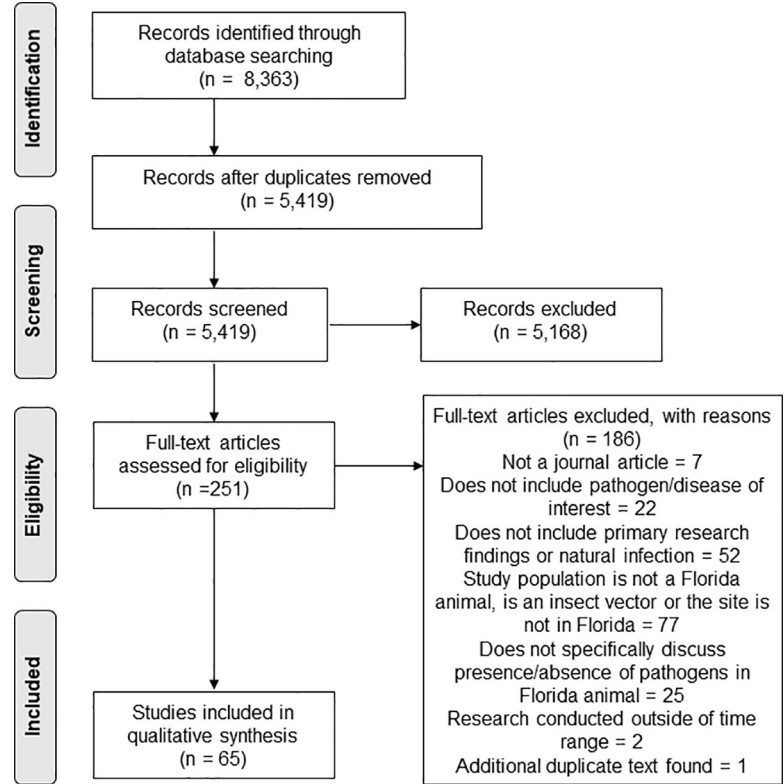

**Fig 1. Study selection flowchart with screening process for inclusion in systematic review.**

OR Whitmore* OR "Burkholderia pseudomallei" OR "Neurotoxic Shellfish Poisoning" OR Brevetoxin* OR Polio* OR "Saint Louis Encephalitis" OR Salmonell* OR Saxitoxin OR "Paralytic Shellfish" OR Shigell* OR Tularemia OR "Typhoid" OR "Venezuelan Equine Encephalitis" OR Vibrio* OR "Viral Hemorrhagic" OR "Viral Haemorrhagic" OR "West Nile" OR "Yellow Fever" OR Zika)

b. AND (Florid* OR "Southeastern United States")

A full list of the search strings designed for each database, including the total number of results returned for each, is available (S2 File). Publications were restricted to peer-reviewed journal articles written in English and published between January 1, 1999 and December 31, 2019. Articles were determined to be eligible if they contained primary study findings documenting a naturally-derived water-related pathogen of interest in animals or animal samples collected in Florida, USA during the same time frame. Such primary findings would include, for example, samples or specimens collected directly from live or deceased animals that are native to Florida with analysis conducted by the author(s). Secondary data analysis, reviews, and modeling only papers were excluded. The search was conducted between April 2nd and April 8th, 2020.

## Screening process

Results from each database search were saved using a citation management program. These results were then copied into a master folder and three researchers removed duplicates by

hand to reach unanimity. The non-duplicate titles were then screened to search each available title and abstract to retain publications that were: a) a journal article; b) written in English due to co-author language limitations; c) primary studies on animal samples collected in Florida; d) primary studies on a pathogen or toxin of interest; and e) naturally occurring contamination or infection. Journal articles that remained went through a full-text review using the same parameters. At least two researchers read the full-text of each article to determine inclusion or exclusion and when necessary, the third researcher was the tie-breaker.

## Results

The initial search resulted in 8,363 articles across the multiple databases (Fig 1). Of these, 2,944 were duplicate titles and were removed. From the remaining 5,419 articles, 251 met the necessary criteria for a full-text review, based on title and abstract screening. After assessment of each full-text title by at least two of the co-authors, 186 articles were removed based on predetermined exclusion criteria. These titles were: a publication type other than peer-reviewed journal article, research not on a pathogen or toxin of interest, findings from experimental infection or did not demonstrate prevalence data in a Florida animal population, study samples were collected outside of the date range, or an additional duplicate article. This resulted in 65 articles accepted for analysis in this review.

The included studies spanned the full date range from 1999–2019 with the majority of publications in the year 2009 (n = 7; Table 2). Study sites were located across the state of Florida with the bulk identified in Brevard county (n = 10), followed by Indian River county (n = 9), Volusia county (n = 8), and Sarasota county (n = 7; Fig 2). Research topics included: disease prevalence study in an animal population, case report or case series on *animal* morbidity or mortality, case report or case series on *human* morbidity or mortality associated with animal contact, or sentinel surveillance for vector-borne pathogens. While most studies were specifically focused on waterborne disease(s), water-related vector-borne disease(s) or water-based toxin(s), several studies investigated multiple categories at the same time (Fig 2A). There was also diversity in the types of animal species sampled across the studies (Fig 2B). Animal categories included: marine and aquatic species; wildlife, reptiles, and birds; and livestock, poultry and companion animals.

### Pathogens and toxins by category

While the search criteria included many waterborne pathogens, water-related vector-borne pathogens and water-based toxins, fewer studies focused on vector-borne agents (n = 12) and toxins (n = 33) than waterborne diseases (n = 39). Waterborne pathogens included: *Escherichia coli* (*E. coli*), *Salmonella* spp., *Cryptosporidium* spp., *Giardia* spp., *Leptospira* spp., *Campylobacter* spp., and *Vibrio* spp. Water-related vector-borne pathogens included: Venezuelan equine encephalitis virus (VEEV), Eastern equine encephalitis (EEE), Western equine encephalitis, West Nile virus (WNV), and St. Louis encephalitis irus (SLEV). Water toxins included: brevetoxins produced by *Karenia brevis (K. brevis)* and toxins saxitoxin, tetrodotoxin, and ciguatoxin. The most frequently examined pathogens and toxins overall included: Toxin-producer *K. brevis* (n = 17), West Nile virus (n = 9), and *Cryptosporidium spp.*, *Giardia spp.*, and *Vibrio spp.* (each n = 8).

### Marine and aquatic animal studies

The majority of study results centered on diseases of interest in marine and aquatic animal populations (n = 35). These animal populations included: wild fish (n = 10), dolphins (n = 9), manatees (n = 5), oysters (n = 5), sea turtles (n = 2), clams (n = 4), other bivalves or snails

**Table 2. Summary of studies from 1999–2019 that found waterborne disease, water-related vector-borne disease, and water-based toxins and toxin-producers of public health importance in Florida animals (n = 65) by positive samples only.**

| Reference | Animal Population | Pathogen, Toxin, or Toxin-Producer Analyzed and Animal(s) Positive n (%) (If Specified) | Florida County (If Specified) |
|---|---|---|---|
| **Marine and Aquatic Studies** | | | |
| Poli et al., 1999 [14] | Clams | • *Karenia brevis* | • Sarasota<br>• Manatee |
| | Whelks | | |
| Trainer & Baden, 1999 [15] | Manatees | • *Karenia brevis (formerly known as Gymnodinium breve)*<br>• Saxitoxin | - |
| Ellison et al., 2001 [16] | Oysters | • *Vibrio parahaemolyticus* | • Alachua |
| Centers for Disease Control and Prevention, 2002 [17] | Pufferfish | • Saxitoxin | • Brevard |
| Fayer et al., 2003 [18] | Oysters | • *Cryptosporidium* spp.<br> ○ 5 (20%) | • Indian River<br>• Brevard |
| | Clams | • *Cryptosporidium* spp.<br> ○ 25 (4%) | • Volusia<br>• St. Johns |
| Buck et al., 2006 [19] | Dolphins | • *Vibrio alginolyticus*<br> ○ 189 (70%)<br>• *Vibrio damsela*<br> ○ 189 (64%)<br>• *Vibrio fluvialis*<br> ○ 189 (18%)<br>• *Vibrio furnissi*<br> ○ 189 (7%)<br>• *Vibrio parahaemolyticus*<br> ○ 189 (17%) | • Sarasota<br>• Manatee |
| Landsberg et al., 2006 [20] | Pufferfish | • Saxitoxin | • Indian River<br>• Brevard<br>• Volusia |
| Greig et al., 2007 [21] | Dolphins | • *E. coli*<br> ○ 33 (48%) | • St. Lucie<br>• Indian River<br>• Brevard<br>• Volusia |
| Naar et al., 2007 [22] | Wild Fish | • *Karenia brevis* | • Gulf |
| Fire et al., 2008 [23] | Dolphins | • *Karenia brevis*<br> ○ 30 (63%) | • Sarasota<br>• Manatee |
| Fire et al., 2008 [24] | Wild Fish | • *Karenia brevis* | • Sarasota<br>• Manatee |
| Pierce & Henry, 2008 [25] | Oysters | • *Karenia brevis* | • Sarasota<br>• Manatee |
| | Clams | | |
| Deeds et al., 2008 [26] | Pufferfish | • Saxitoxin<br>• Tetrodotoxin | • Indian River<br>• Brevard<br>• Volusia |
| Abbott et al., 2009 [27] | Pufferfish | • Saxitoxin | - |
| Schaefer et al., 2009 [28] | Dolphins | • *Vibrio alginolyticus*<br>• *E. coli*<br> ○ –(9.3%) | • Indian River<br>• Brevard<br>• Volusia |
| Schaefer et al., 2009 [29] | Dolphins | • VEEV<br> ○ 115 (8.7%)<br>• EEE<br> ○ 116 (1.7%)<br>• WNV<br> ○ 118 (4.24%)<br>• WEE<br> ○ 116 (3.34%) | • Indian River<br>• Brevard<br>• Volusia |
| Nam et al., 2010 [30] | Lemon Sharks | • *Karenia brevis* | • Brevard |
| Wetzel et al., 2010 [31] | Manatees | • *Karenia brevis* | - |

*(Continued)*

**Table 2.** (*Continued*)

| Reference | Animal Population | Pathogen, Toxin, or Toxin-Producer Analyzed and Animal(s) Positive n (%) (If Specified) | Florida County (If Specified) |
|---|---|---|---|
| Morris et al., 2011 [32] | Dolphins | • *V. alginolyticus*<br>• *E. coli* | • Indian River<br>• Brevard<br>• Volusia |
| Twiner et al., 2011 [33] | Dolphins | • *Karenia brevis* | • Sarasota<br>• Manatee |
| | Snails | | |
| | Wild Fish | | |
| Twiner et al., 2012 [34] | Dolphins | • *Karenia brevis*<br> ○ - (52%) | • Gulf |
| van Deventer et al., 2012 [35] | Wild Fish | • *Karenia brevis*<br> ○ 12 (100%) | • Sarasota |
| Capper et al., 2013 [36] | Manatees | • *Karenia brevis* and/or Saxitoxin<br> ○ 14 (64%) | • Pinellas |
| | Green Sea Turtles | • *Karenia brevis* and/or Saxitoxin<br> ○ 13 (85%) | |
| Staley et al., 2013 [37] | Oysters | • *Vibrio spp.* | • Hillsborough |
| Unger et al., 2014 [38] | Black Tip Sharks | • *Vibrio alginolyticus* | • Martin<br>• Palm Beach |
| McFarland et al., 2015 [39] | Green Mussels | • *Karenia brevis* | • Lee |
| Walsh et al., 2015 [40] | Manatees | • *Karenia brevis* | • Citrus |
| Perrault et al., 2016 [41] | Loggerhead Sea Turtles | • *Karenia brevis*<br> ○ 34 (100%) | • Monroe |
| Vorbach et al., 2017 [42] | Manatee | • *Salmonella* spp<br> ○ 1 (100%) | • Martin |
| Kemp et al., 2017 [43] | Coral | • *Vibrio spp.* | • Monroe |
| Hardison et al., 2018 [44] | Lionfish | • Ciguatoxin<br> ○ 20 (25%) | • Monroe |
| Martony et al., 2018 [45] | Sea Urchins | • *Vibrio spp.*<br> ○ 70 (33%) | • Monroe |
| Rolton et al., 2018 [46] | Clams | • *Karenia brevis* | • Lee |
| Fire et al., 2019 [47] | Dolphins | • *Karenia brevis*<br> ○ 119 (12%)<br>• Saxitoxin<br> ○ 119 (13%) | • Indian River<br>• Brevard<br>• Volusia |
| Fang et al., 2019 [48] | Oysters | • *Vibrio cholerae*<br> ○ 60 (48%)<br>• *Vibrio parahaemolyticus*<br> ○ 60 (100%)<br>• *Vibrio vulnificus*<br> ○ 60 (100%) | • Franklin |
| | Wild Fish | • *Vibrio parahaemolyticus*<br> ○ 38 (83%)<br>• *Vibrio vulnificus*<br> ○ 38 (67%) | |
| **Terrestrial Wildlife, Reptile, and Bird Studies** | | | |
| Hudson et al., 2000 [49] | Wild Birds | • *Salmonella* spp. | - |
| Blackmore et al., 2003 [50] | Wild Birds | • WNV<br> ○ 7,669 (14.4%) | - |
| Jacobson et al., 2005 [51] | Alligators | • WNV<br> ○ 3 (100%) | - |
| Godsey et al., 2005 [52] | Wild Birds | • WNV<br> ○ 152 (10.5%)<br>• SLEV<br> ○ 152 (1.3%) | • Jefferson |

(*Continued*)

**Table 2.** (*Continued*)

| Reference | Animal Population | Pathogen, Toxin, or Toxin-Producer Analyzed and Animal(s) Positive n (%) (If Specified) | Florida County (If Specified) |
|---|---|---|---|
| Allison et al., 2005 [53] | Double Crested Cormorants | • WNV<br>  ○ 2 (50%) | • Monroe |
| Gibbs et al., 2006 [54] | Feral Pigs | • WNV<br>  ○ 35 (17%) | - |
| Nemeth et al., 2009 [55] | Crested Caracaras | • SLEV<br>  ○ 80 (3%)<br>• EEE<br>  ○ 80 (3%)<br>• WNV<br>  ○ 80 (9%) | • Collier<br>• DeSoto<br>• Hardee<br>• Hendry<br>• Highlands<br>• Martin |
| Charles-Smith et al., 2009 [56] | Gopher Tortoises | • *Salmonella* spp.<br>  ○ 60 (10%) | • St. Johns<br>• Orange |
| Landolfi et al., 2010 [57] | Green Snakes | • *Cryptosporidium* spp.<br>  ○ 5 (100%). | - |
| van Deventer et al., 2012 [35] | Shorebirds | • *Karenia brevis*<br>  ○ 16 (100%) | • Pinellas<br>• Lee<br>• Charlotte<br>• Sarasota |
| Chatfield et al., 2013 [58] | Feral Pigs | • *Leptospira* spp.<br>  ○ 324 (33%) | - |
| Grigione et al., 2014 [59] | Coyotes | • *Cryptosporidium* spp.<br>  ○ 40 (13%)<br>• *Giardia canis*<br>  ○ 40 (8%) | • Pinellas |
| Hernandez et al., 2016 [60] | White Ibis | • *Salmonella* spp.<br>  ○ 333 (16.5%) | • Palm Beach<br>• Broward<br>• Miami-Dade |
| Huffman et al., 2018 [61] | Gopher Tortoises | • *Cryptosporidium* spp.<br>  ○ 123 (1%) | • Indian River<br>• Martin<br>• Palm Beach |
| **Livestock and Companion Animal Studies** | | | |
| Blackmore et al., 2003 [50] | Chickens | • WNV<br>  ○ 2,128 (9%) | - |
| | Horses | • WNV<br>  ○ 948 (52%) | - |
| Riley et al., 2003 [62] | Beef Cattle | • *E. coli*<br>  ○ 296 (9%) | • Hernando |
| Trout et al., 2004 [63] | Dairy Cattle | • *Giardia duodenalis* | - |
| Godsey et al., 2005 [52] | Domestic Birds | • WNV<br>  ○ 201 (11.4%)<br>• SLEV<br>  ○ 201 (1%) | • Jefferson |
| Trout et al., 2005 [64] | Dairy Cattle | • *Giardia duodenalis* | - |
| Centers for Disease Control and Prevention, 2005 [65] | Cattle<br>Sheep<br>Goats | • *E. coli*<br>  ○ 36 (16.7%) | - |
| Trout et al., 2006 [66] | Dairy Cattle | • *Giardia duodenalis*<br>  ○ 86 (24.4%) | - |
| Coffey et al., 2006 [67] | Domestic Dogs | • VEEV<br>  ○ 633 (4%) | - |
| Trout et al., 2007 [68] | Dairy Cattle | • *Giardia duodenalis* | - |
| Fayer et al., 2007 [69] | Dairy Cattle | • *Cryptosporidium* spp.<br>  ○ 93 (5.4%) | - |

(*Continued*)

**Table 2.** (Continued)

| Reference | Animal Population | Pathogen, Toxin, or Toxin-Producer Analyzed and Animal(s) Positive n (%) (If Specified) | Florida County (If Specified) |
|---|---|---|---|
| Riley et al., 2008 [70] | Beef Cattle | • *Salmonella* spp.<br>• *Campylobacter* spp. | • Hernando |
| Rios et al., 2009 [71] | Horses | • WNV<br>  ○ 936 (57%) | - |
| Alelis et al., 2009 [72] | Goats | • *E. coli* | • Pinellas |
| Sabshin et al., 2012 [73] | Domestic Cats | • *Cryptosporidium* spp.<br>  ○ 100 (15%)<br>• *Giardia* spp.<br>  ○ 100 (14%)<br>• *Salmonella* spp.<br>  ○ 100 (5%) | • Alachua |
| Tupler et al., 2012 [74] | Domestic Dogs | • *Cryptosporidium* spp.<br>  ○ 100 (7%)<br>• *Giardia* spp.<br>  ○ 100 (19%)<br>• *Salmonella* spp.<br>  ○ 100 (4%) | • Alachua |
| Estep et al., 2013 [75] | Chickens | • EEE | • Walton |
| Wyrosdick et al., 2017 [76] | Domestic Cats | • *Cryptosporidium* spp.<br>• 76 (6.6)<br>• *Giardia* spp.<br>• 76 (3.9) | • Citrus |
| Montgomery et al., 2018 [77] | Domestic Dogs | • *Campylobacter* spp. | - |
| Baker et al., 2019 [78] | Dairy Cattle | • *E. coli* | - |

Note: WNV = West Nile Virus; EEE = Eastern Equine Encephalitis; VEEV = Venezuelan Equine Encephalitis; SLEV = St. Louis Encephalitis; WEE = Western Equine Encephalitis

(n = 3), sharks (n = 2), sea urchin (n = 1), and coral (n = 1). The most frequent pathogens and toxins found in marine and aquatic animal populations in Florida were toxin-producer *K. brevis* (n = 17), *Vibrio spp.* (n = 8), and saxitoxin (n = 7). The most frequent locations for these studies were the counties of Brevard (n = 10), Indian River (n = 8), Volusia (n = 8), Sarasota (n = 7), and Manatee (n = 6). Most of these studies investigated disease prevalence in a marine animal population, although not as part of any sentinel surveillance program for public health.

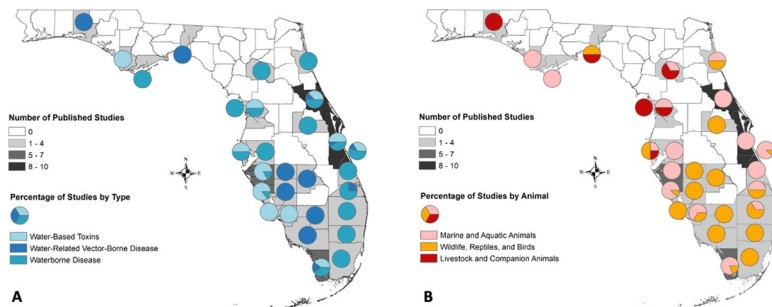

**Fig 2.** Density of the number of published studies by Florida county and the percentage of studies by type of disease or toxin (A) and category of animals sampled (B). Base boundary map of Florida counties reprinted under a CC BY license, with permission from the Southwest Florida Water Management District, original copyright 2009.

### Terrestrial wildlife, reptile and bird studies

Publications were also found regarding land animal populations including reptile and bird species (n = 14). These animal populations included: various wild birds (n = 7), feral pigs (n = 2), gopher tortoises (n = 2), coyotes (n = 1), green snakes (n = 1), and alligators (n = 1). The most frequent pathogens found in terrestrial animal and wild bird populations included: West Nile Virus (n = 6), *Cryptosporidium* spp. (n = 3), *Salmonella* spp. (n = 3). The most frequent locations for these studies named throughout the state of Florida were Pinellas county (n = 2), Martin county (n = 2), and Palm Beach county (n = 2). These studies investigated disease prevalence in land animal populations as well as cases regarding land animal morbidity and mortality as a result of diseases of interest.

### Livestock and companion animal studies

Studies were also identified in the search that investigated diseases of interest in livestock (including poultry) and companion animals (n = 19). These animal populations included: cattle (n = 9), companion dogs (n = 3), companion cats (n = 2), chickens (n = 2), goats (n = 2), and horses (n = 2), and other domestic or captive birds (n = 1). The most frequent pathogens found in these domestic animal populations were: *Giardia* spp. (n = 7), *E. coli* (n = 4), *Cryptosporidium* spp. (n = 4), and West Nile virus (n = 4). Many of these studies regarding domestic animal populations did not specify a Florida county where data collection occurred (n = 12). Of those with a reported study site, the most frequent locations were Hernando county (n = 2) and Alachua county (n = 2). Several of the results in livestock and companion animal populations were from investigations into the prevalence of waterborne and sanitation-related diseases. among livestock populations such as beef and dairy cattle throughout the state of Florida.

## Discussion

Between 1999 and 2019, sixteen water-related pathogens of human health importance were documented in Florida's animal populations. Of the included studies in this review, most focused on zoonotic waterborne diseases such as *E. coli*, *Salmonella* spp., *Cryptosporidium* spp., *Giardia* spp., *Leptospira* spp., *Vibrio* spp., and *Campylobacter* spp. [16, 18, 19, 21, 28, 32, 37, 38, 42, 43, 45, 48, 49, 56–66, 68–70, 72–74, 76–78]. Waterborne diseases are associated with agricultural runoff, stormwater and flooding, trash, and poorly managed sewage [79–81]. Fecal shedding of zoonotic enteric pathogens can pollute natural habitats and water systems which can lead to human exposure [82, 83].

### Waterborne disease

*Escherichia coli* (*E. coli)* was reported in marine mammal populations such as dolphins and manatees, as well as in livestock populations like cattle and goats, typically in their fecal samples [21, 28, 32, 62, 65, 70, 72, 78]. While *E. coli* is a broad species of bacterial subtypes, most of which is harmless and naturally occurring in human and animal intestinal flora, at least six pathotypes are known as diarrheagenic *E. coli* and can cause disease [84]. The FDOH has included the STEC pathotype, or Shiga toxin-producing *E. coli*, as one of the reportable waterborne diseases in Florida. The larger subset of *E. coli* studies included in this analysis did not indicate the subtype detected and therefore the pathogenicity could not be determined. However, the presence of *E. coli* as a fecal indicator may demonstrate proximal waste contamination and exposure potential for other waterborne zoonotic diseases. Confirmed outbreaks of *E. coli* in this study were often associated with environments with high levels of human and

animal interaction, such as petting zoos and agricultural areas [62, 65, 70, 72, 78]. The indiscriminate water-focused fecal shedding of zoonotic pathogens, such as pathogenic *E. coli*, from marine mammals is a point of concern since the state thrives on water-based tourism and recreation. There is a potential risk of humans encountering a diseased animal or their feces in or near the beach environments [85]. Waterborne diseases. have been found in multiple marine mammals and shorebirds, including seals that spend significant time in shared spaces with beachgoers [85–87]. Further research into waterborne disease transmission and exposure risks in marine animals and humans in Florida is warranted.

*Cryptosporidium* spp. was primarily found in terrestrial animals such as coyotes and tortoises but was also associated with outbreaks in livestock and companion animals (i.e. shelter animals) [57, 59, 61, 69, 73, 74, 76]. Animals in veterinary care, shelters, pet stores, petting zoos or wildlife rehabilitation centers may harbor disease [88–91]. Because of the high level of human contact associated with animal shelters and wildlife rehabilitation centers, it is important to monitor the ongoing health of these animals. If an animal is infected with a disease of significance to human or veterinary health, practitioners should determine whether the animal had the disease prior to their stay, contracted it because of unhygienic containment conditions, or developed it after exposure to another infected animal or human handler [92]. Employees and animal health providers should maintain a high level of hygiene and health screenings for animals in their care as well as their human visitors [89, 90, 93]. Any rehomed, purchased, traded, or released animal should be healthy and disease-free to avoid further transmission to the environment [85, 91, 92].

Livestock are a prevalent source of water and foodborne illness, especially when animal waste is not safely managed and personal hygiene measures are not adhered [79, 80, 88]. Florida houses approximately 47,000 commercial farms and ranches [94]. As agricultural animals and animal products are transported in and out of Florida, preventing and responding to zoonotic waterborne disease can hinder the potential economic and herd loss associated and impede multi-state outbreaks [95]. Transporting live animals who may be under duress due to high-volume agricultural facilities can increase the risk for transmission of waterborne disease within herds and to humans [96]. Care and concern should be taken for the conditions of livestock food, water, and confinement areas in order to prevent potential contamination [88, 96]. Many of our results pertaining to water-related pathogens found in Florida livestock populations did not disclose which counties the data were collected, most likely to protect the agricultural facility from negative consumer opinion [50, 67, 71, 97]. It is important to mitigate the risk of transmission to animals and humans by implementing preventive measures and continuing monitoring and surveillance in the environment. Ensuring a safe built environment and habitat for domestic animals can help mitigate the risk of illness for animals and people [71].

Companion animals also have frequent interactions and close contact with humans of all ages [98]. People who may be immunocompromised or otherwise at risk for severe morbidity due to a water-related disease may be unaware of the risk that unsafe contact with their pet may pose [93]. Vaccinating animals that have frequent contact with humans, cleaning pet areas and items regularly, washing hands after contact with pets or pet waste, monitoring the health of pets and household members with routine veterinary care, and feeding animals safe feed are some of the ways to prevent zoonotic transmission [99]. Pet animals who freely roam outside may be at risk for contact with infected companion or wild animals, who can transmit pathogens via water, feces, or direct contact [100]. This is especially true in areas where urban and natural landscapes overlap, as wild animals such as coyotes have been shown to pass waterborne disease to companion and domestic animals [59].

## Water-based toxins and toxin-producers

Water-based toxins included brevetoxin produced by overgrowth of *Karenia brevis* and the toxins saxitoxin, tetrodotoxin, ciguatoxin, and others. The most prevalent water-based toxin in the studies analyzed was brevetoxin produced by *K. brevis*, which is associated with harmful algal blooms (HABs), known as "red tides", that can plague the coastline of Florida [14, 15, 22–25, 30, 31, 33, 34, 36, 39–41, 46, 47, 101, 102]. Red tide events have led to die-offs in fish, turtles, dolphins, and manatees while cyanobacteria or blue-green algae has killed birds, wildlife, livestock, dogs and cats [103]. During Florida's red tides or other HAB events, many species of marine animals and shellfish are at risk of being affected by toxin-producer *K. brevis* and other paralytic toxins, ciguatoxins, tetrodotoxins, brevetoxins, which can contribute to human exposure through consumption of contaminated seafood or contact with the toxins through water or aerosolized particles [104, 105]. It is important to remain aware of red tide seasons, inform the public, and exercise caution when entering affected areas or bringing pets or livestock near water presenting with a harmful algal bloom [102, 103].

Most of the studies included in our analysis were conducted in marine and aquatic animal populations and concentrated in South Florida, especially the Sarasota Bay and the Indian River Lagoon area [14, 18–21, 23–26, 28, 29, 32, 33, 47]. Water-based toxins were the most frequently documented disease-causing agent in this group [14, 15, 17, 20, 22–27, 30, 31, 33, 34, 36, 39–41, 44, 46–48]. Research and prevention efforts to curb HABs and other water toxin effects in Florida and along its coast are ongoing and must consider the dangers these organisms present to human, environmental and animal health. Collaborative efforts such as the Bottlenose Dolphin HERA (Health and Risk Assessment) Project have been instrumental in fostering interdisciplinary research efforts for marine mammals [28]. HAB surveillance and response efforts in Florida are a joint One Health collaboration between the Florida Department of Health, Florida's Poison Control Centers and hospitals, the National Ocean and Atmospheric Administration (NOAA), Florida Fish and Wildlife Conservation Commission, Mote Marine's Beach Conditions Report, the Florida Department of Environmental Protection, Fish and Wildlife Research Institute, the Centers for Disease Control and Prevention, Florida Water Management Districts, and the Gulf of Mexico Coastal Ocean Observing System [106]. There are opportunities for further research on water-based toxins in marine animal populations of Florida with priority given to regions with warmer air and water temperatures and greater risk for human-animal contact or exposure to shared water environments.

Human cases of Neurotoxic and Paralytic Shellfish Poisoning are most often caused by the consumption of shellfish and other seafood that contains water-based toxins, typically sourced from contaminated water bodies [104]. Animal cases of algal toxin poisoning can be the result of direct contact in the water or subsequent grooming behaviors, inhalation of aerosolized particles, and bioaccumulation in the foodweb [105]. Water-based toxins and their resulting HABs can be found in freshwater, saltwater, and within a mix of salt water and freshwater (brackish water) [107]. In South and Southwest Florida, a majority of the natural water environments are considered brackish, both a natural condition as coastal waterways originate from inland freshwater springs that flow out to the saltwater ocean and a result of saltwater intrusion [108]. These specialized marine ecosystems are ideal habitats for commercially harvested fish and shellfish. However, the unique salinity and warm temperatures of the waterways may also serve to proliferate ciguatoxins, which can result in Ciguatera fish poisoning [109].

## Water-related vector-borne disease

Water-related vector-borne diseases Venezuelan Equine Encephalitis Virus (VEEV), Eastern Equine Encephalitis (EEE), West Nile Virus (WNV), and St. Louis Encephalitis Virus (SLEV)

were each documented in Florida animals in this search. Publications included in our study were largely concerned with bird populations such as wild birds and Crested Caracaras due to their migratory patterns and characteristics as effective hosts for arboviral diseases [52, 55, 75]. However, evidence of arboviral infection was also found in chicken, horse, and alligator populations [50, 51, 54, 67]. Sentinel surveillance is an important tool in Florida used to monitor the occurrence of water-related vector-borne disease prevalence in chickens, horses, and mosquitoes collected and tested in different counties across the state [110]. In particular, sentinel surveillance using chickens has been a successful method for monitoring the presence of emerging or reemerging arboviral diseases and to provide early public health warnings [110]. While WNV can cause bird die-offs among corvids, chickens do not effectively amplify arboviruses and are not a considered a risk to people for mosquito-borne viruses [110]. Like chickens, horses are also an important sentinel species to indicate rates of water-related vector-borne and arboviral diseases, such as EEE and WNV, circulating in an area however they tend to show more signs and symptoms of infection [110].

The climate of Florida with higher temperatures and precipitation coupled with abundance water bodies provide excellent environments for water-related disease vector (i.e. mosquitoes) to proliferate [111, 112]. Between 1999 and 2019, Florida maintained a yearly average temperature of 71.5˚F with 52.9 inches of rain [113]. Expected climatic changes such as increased temperatures, flooding and extreme weather events, and rainfall over the coming years will mean a need for increased monitoring and sentinel surveillance for arboviral and vector-borne diseases and their vectors [112]. Monitoring the health of wild birds is also vital to determine the disease burdens of migratory species which serve as hosts for arboviral disease [114]. Research is still needed on vector-borne pathogen trends and seasonality among wild birds and the effects of avian amplification of these diseases in Florida [114–116]. Avian amplification is a phenomenon in which co-existence of mosquito vectors and avian hosts can increase the risk of transmission of important arboviral diseases [115].

**One health recommendations.**  Surveillance efforts should expand to a greater diversity of animals to better meet the needs of inland regions of Florida, where loss of natural wildlife habitats due to human development and climate change is likely to increase the risk of contact and transmission with vector hosts and disease reservoirs [7, 117]. Human and terrestrial animal interactions occur either directly or by using shared environments and freshwater sources, which can result in increased risk of transmission of zoonotic diseases [118]. Future efforts should advocate for increased disease surveillance and protection of Florida's diverse avian, marine and terrestrial animal populations and their natural range through a One Health coordinated approach [118, 119]. New research could spur novel findings on water-related disease and transmission risks in these populations, while also advocating for the protection and preservation of the environments in which they live. More research on safe and healthy coexistence with natural wildlife is needed, both today and in the future [59, 115, 120].

Using a One Health approach to simultaneously strengthen the safety and well-being of humans, animals and the environment is prudent in Florida. Research surrounding water-borne, water-based, and water-related diseases and toxins that affect human and animal populations in Florida should take advantage of opportunities for collaboration geared towards prevention efforts and education of the public. This holistic approach could also be considered when designing a study for these water-related pathogens and toxins as resources such as field staff, sampling and laboratory equipment, and even specimens could be multi-purpose by investigating the disease-causing agents from animal, human, and environmental angles with joint teams of professionals from a variety of fields and public health interests. Sharing datasets, locations for collection, research findings and funding opportunities between human and veterinary public health teams is the best way to move forward under a united approach to

conserving and protecting Florida's health. In addition, researchers who are publishing their findings in public health, veterinary, wildlife, and zoological journals would benefit a wider-ranging audience if they consider how the presentation of their data and study results could be utilized by colleagues in complementary fields.

Further research on a wider variety of water-related pathogens and toxins in more animal populations can help inform policy and health promotion efforts. The research presented by this study demonstrates regions of the state which are lacking information related to these pathogens and toxins of public and veterinary importance. For example, while Brevard county had many published studies on aquatic and marine animal populations, no studies were done to examine these disease agents in terrestrial wildlife, birds, reptiles, or domestic animals or livestock. The interest in marine mammal health assessments and publications may correspond with local research programs and centers within the counties such as the Hubbs Sea-world Research Institute in Brevard county or the Sarasota Dolphin Research Program in Sarasota county. However, the maps of the published studies also show that single studies have been done for many counties while large portions of the state do not have any works from the past twenty years, particularly in regions that lack university connections or research centers. Yet disease surveillance efforts by county health departments track and record each reported human case of these conditions and the FDOH has created a public repository of these frequencies through the website FLHealthCHARTS (http://www.flhealthcharts.com/charts/default.aspx). Animal prevalence data could be shared in a similar manner so that zoonotic diseases and toxins could be searched by several health response or research teams. Information sharing should be encouraged across all counties in Florida and facilitated through formal surveillance channels that are open to experts and field professionals from multiple sectors. It is important to consider input and expertise from all silos in order to best serve our Florida communities.

This study is not without its limitations. Utilizing search databases more commonly available to public health researchers may have unintentionally excluded discipline-specific animal science, veterinary, ecology, or zoology journals. These journals may have published research which met our criteria and would have resulted in an even greater understanding of water-related pathogen and toxin presence in Florida's animals. Further research on this topic is warranted to investigate potential risk associations of water-related pathogens and toxins as they pertain to Florida's animals, as this objective falls outside the scope of this systematic review. Moreover, without *E. coli* pathotypes provided by many of the included studies, the role of this microorganism as a zoonotic disease risk could not be fully assessed. Due to the language limitations of the co-authors, journal articles written in any language other than English were also excluded, which could have meant a loss in key findings. And finally, as with many systematic review screenings conducted by hand, there is room for human error. In assessing over 8,000 articles for duplication and adherence to inclusion criteria, it is possible that a publication was misidentified which should have been analyzed in the final results. The researchers attempted to combat these potential errors with regular laboratory team meetings and at-length discussion on the merits of each titles for inclusion.

## Conclusion

Florida's environmental conditions present a unique opportunity for water-related pathogens and toxins to spread but also provides the chance to conduct collaborative research on how to prevent infection in humans and animals. This review highlights what water-related diseases of public health importance have been found over the past twenty years among animals in Florida but also demonstrates where research is lacking for specific pathogens and aquatic

toxins, regions, and animal populations. Prevention and protection efforts for Florida's people, animals, and environment necessitates a One Health approach to tackle waterborne disease, water-related vector-borne disease, and water-bases toxins in a successful, interdisciplinary manner.

## Supporting information

**S1 File. PRISMA checklist.**
(DOC)

**S2 File. Search strings and results by database.**
(DOC)

## Acknowledgments

The authors would like to thank Mehgan Fox for her excellent research assistance and additional partner members of the Coastal One Health and Zoonoses (COHZ) lab.

## Author Contributions

**Conceptualization:** Meg Jenkins, Sabrina Ahmed, Amber N. Barnes.

**Data curation:** Meg Jenkins, Sabrina Ahmed, Amber N. Barnes.

**Formal analysis:** Meg Jenkins, Sabrina Ahmed, Amber N. Barnes.

**Funding acquisition:** Amber N. Barnes.

**Methodology:** Amber N. Barnes.

**Project administration:** Meg Jenkins, Amber N. Barnes.

**Resources:** Meg Jenkins, Sabrina Ahmed, Amber N. Barnes.

**Supervision:** Amber N. Barnes.

**Visualization:** Meg Jenkins, Sabrina Ahmed.

**Writing – original draft:** Meg Jenkins, Sabrina Ahmed, Amber N. Barnes.

**Writing – review & editing:** Meg Jenkins, Sabrina Ahmed, Amber N. Barnes.

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
