## [Decision Letter · Decision Letter 0]

17 Feb 2021

PONE-D-20-32325

A Systematic Review of Waterborne and Water-Related Disease in Animal Populations of Florida from 1999-2019

PLOS ONE

Dear Dr. Barnes,

Thank you for submitting your manuscript to PLOS ONE. After careful consideration, we feel that it has merit but does not fully meet PLOS ONE’s publication criteria as it currently stands. Therefore, we invite you to submit a revised version of the manuscript that addresses the points raised during the review process.

The reviewers agree that the spatial and temporal scope of the review is relevant and well done and both reviewers also agree that this review will be useful. The technical aspects of the review are also of quality.  It has been pointed out that no new conclusions have been generated from the review.  Including these new insights will bring this manuscript in line with the PLOS ONE criteria for systematic reviews.  Furthermore, one reviewer suggests including a map or other figure grouping the studies by animal/disease/location, and I agree this would be beneficial. 

We look forward to receiving your revised manuscript.

Kind regards,

Brett Austin Froelich, Ph.D

Academic Editor

PLOS ONE

Journal Requirements:

Additional Editor Comments:

Thank you for your submission to PLOS ONE. PLOS ONE does not consider reviews and thus this submission is not appropriate for this journal.

Reviewers' comments:

Reviewer's Responses to Questions

**Comments to the Author**

1. Is the manuscript technically sound, and do the data support the conclusions?

Reviewer #1: Yes

Reviewer #2: Yes

2. Has the statistical analysis been performed appropriately and rigorously? 

Reviewer #1: N/A

Reviewer #2: N/A

3. Have the authors made all data underlying the findings in their manuscript fully available?

Reviewer #1: Yes

Reviewer #2: Yes

4. Is the manuscript presented in an intelligible fashion and written in standard English?

Reviewer #1: Yes

Reviewer #2: Yes

5. Review Comments to the Author

Reviewer #1: Reviewer comments to authors also uploaded as an attachment.

The study entitled “A Systematic Review of Waterborne and Water-Related Disease in Animal Populations of Florida from 1999-2019” provides a summary of reports of animal diseases with public health relevance in Florida based on a systematic review of primary scientific literature. This study is timely as animals are increasingly recognized as an important reservoir for human disease agents.

The authors report on the numbers of studies associated with various types of animals, various types of disease agents, and locations (counties) in Florida. However, they do not report or analyze any quantitative data from the studies of interest, and are thus unable to draw conclusions about the risk or prevalence of various animal-associated human disease agents in Florida. It is unclear if or how the number of papers reporting on a given animal-associated disease agent correlates with meaningful information about animal-associated disease risks in the state. Despite this, it may be useful for vets, health officials, policy makers, and scientists to have this summary of research done on various animals and disease agents in Florida. To this end, the paper would benefit from a map, table and/or figure summarizing included studies by animal type, disease agent, and location.

Synthesizing the information from their review would also help the authors meet PLOS One’s criteria for systematic reviews, which states that the journal accepts systematic reviews that use “explicit, systematic methods to identify, select, and critically appraise relevant research, and to collect and analyze data from the studies that are included in the review”. While the authors here systematically identified the studies they have summarized, they did not present new ideas, analyses, or conclusions based on the published data. This manuscript, particularly the discussion, reads more as a traditional review.

Throughout the manuscript, the authors should clarify and differentiate the different types of disease agents they’re interested in. There are many instances where toxins are grouped in with zoonotic pathogens, and these are very different types of public health risks. Specific instances of where toxins and pathogens should be differentiated are indicated below in the specific comments. In addition, it is not clear from the text how various types of organisms are associated with animals and humans. For example, K. brevis is not itself a pathogen or toxin, but it produces toxins that bioaccumulate in seafood and is dangerous to humans when ingested. The various types of disease agents and links between human illness, animals, and the environment should perhaps be better described.

Though not grammatically or factually incorrect, some of the text is written in a style that is not typical of scientific manuscripts. Several specific examples where extra words or descriptors are used and could be removed are noted below, and the study authors should work with the journal’s editors to remove unnecessary text prior to publication.

Specific comments

Abstract

• Lines 26-27: The term “infectious” is implied for pathogens and confusing here because the authors could also be referring to toxins (which are not infectious), suggest removing.

• Lines 27-28: Diseases/illnesses in Florida are reportable to the department of health regardless of whether they’re occurring in residents or in visitors to the state, correct? Suggest rewording to clarify.

• Line 29: Suggest authors amend text to “water-related pathogens and toxins” here, since both toxins and pathogens are discussed.

• Lines 28-30: Suggest that the authors update text to reflect that the systematic review also aimed to understand to what degree various water-related pathogens are associated with various animal types and their risk to public health.

• Suggest including which waterborne diseases and which animals were included in the searches in the abstract; this is covered later in the manuscript but is central to the study’s aims and methods and should be briefly described in the abstract.

• Line 36: What is meant by “original data”? (Presence/absence? Abundance? By any method?)

• Line 38: Large numbers should not be written as text (70 v. seventy).

Introduction

• Lines 57-28: Mention of them eparks seems irrelevant to discussion of Florida’s waters; suggest removing.

• Line 60: It’s distracting and unnecessary to specify that Florida’s residents are humans.

• Line 62: Most of these animals are not “Florida-specific” because they are also found in other areas, suggest removing.

• Paragraph at line 72: Suggest introducing waterborne infectious diseases, water-related vector-borne diseases, and toxins as related but separate categories of disease agents before defining them. In general, authors should be sure that toxins in particular are differentiated from infectious disease agents. It would also be helpful to have examples of each of these disease agent categories explaining animal association link to human disease.

• Line 76: Fix typo “are can be”.

• Line 78: Toxin-producing organisms are not themselves “contaminated” organisms, perhaps the authors meant to that they “contaminate” water bodies? Please clarify.

• Line 86: Run-on sentence, suggest splitting into at least two sentences.

• Line 91: Replace “have been impacting” with “impact”.

Methods

• Paragraph beginning at line 97 and Table 1: Are all of these water-related diseases related to animals? It’s not clear from the text or table legend. Based on the list of disease agents, this appears to be the case, but the authors should clarify. It would also be useful to include information on what types of animals these various diseases are associated with.

• Line 107: Suggest providing a short description of PRISMA guidelines.

• Line 138: What types of original data were included? Suggest providing details and/or examples.

• Lines 145 and 149: Not necessary to specify that articles were screened or reviewed by the coauthors.

• Lines 148-149: Item “d” specifies pathogens, but does this include toxins? Please clarify.

• Line 158-162: Syntax and organization of list is distracting, please re-word.

• Consider moving Figure 1 to methods section.

Results

• Figure 1: Consider specifying on figure or in legend why records were excluded during screening; information is included in the text but would be helpful in the figure.

• Line 175: Suggest updating paragraph header to reflect the fact that both pathogens and toxins are discussed.

• Line 191: K. brevis isn’t a pathogen, it’s a toxin-producing organism. Please clarify.

• Suggest including a map with colors to indicate where in the state the authors are reporting studies on various types of animal disease and what types of pathogens are there. This could help vets and public health officials understand where animal diseases of public health relevance have been reported.

• Suggest creating a table or figure with results broadly summarized/synthesized, reflecting text; for example, it would be helpful to see how many studies were associated with each disease agent, animal type, and location, rather than a list of studies.

Discussion

• Line 231: Add period after citations.

• Line 296: K. brevis isn’t a toxin, it produces toxins (brevetoxins).

• Line 312: Toxins are not pathogens, please revise text.

• In general, the discussion is focused on recommendations to reduce risks associated with different types of animal-associated human disease agents, and broad descriptions of included study results. However, the systematic review was aimed at enumerating reports of animal-associated human disease agents, and it is not clear how the results of the review inform the key points in the discussion. Suggest that the authors revise the discussion with a focus on describing broad trends (animals, diseases, and locations) they found in their literature search, how this information could be utilized by the state and/or vets and public health personnel, and identifying opportunities for additional research.

Reviewer #2: This systematic review of waterborne and water-related diseases covered a host of infectious pathogens, zoonotic diseases, and toxin of public health importance in the State of Florida over the past 20 years. The authors searched using keywords and search terms in 19 relevant databases resulting in 8,000 peer reviewed articles of which 70 studies were included for final analysis. All studies were reviewed independently by two authors and discrepancies in findings reviewed by a third author. The authors elucidated the most common types of animals associated with the diseases of interest were not surprisingly marine mammals, fish and shellfish, wild birds and livestock. Within in this group, the most common diseases were Karena brevis, vibriosis, E. coli, and Salmonellosis. The authors point out that there is a disparity between surveillance and reporting between veterinary professionals and public health officials, and a more universally applicable approach such as OneHealth approach would help decrease this burden.

This manuscript was extremely well written and covered the breadth and scope of the topic in a manner that was both easy to understand and informative. The topics were organized logically, and the findings succinctly presented. The information presented was also current, covering the last 20 years. The discussion included both strengths and limitations of the study design and presented the authors’ conclusion as to best handle the specific areas which are lacking. The authors such a OneHealth approach to help populate missing links in an interdisciplinary manner. The reviewer finds this approach interesting and this is the one aspect of the manuscript which lacks more information, but it also may be beyond the scope of the authors’ original intent.

6. PLOS authors have the option to publish the peer review history of their article (what does this mean?). If published, this will include your full peer review and any attached files.

Reviewer #1: No

Reviewer #2: No

---

## [Author Response · Author response to Decision Letter 0]

19 Apr 2021

Please see the attached document for a point-by-point response to each query or critique provided by the reviewers. We appreciate their time and consideration of this manuscript.

---

## [Decision Letter · Decision Letter 1]

11 Jun 2021

PONE-D-20-32325R1

A Systematic Review of Waterborne and Water-Related Disease in Animal Populations of Florida from 1999-2019

PLOS ONE

Dear Dr. Barnes,

Thank you for submitting your manuscript to PLOS ONE. Many improvement were made to the manuscript.  Only minor revisions remain.  Therefore, we invite you to submit a revised version of the manuscript that addresses the points raised during the review process.

We look forward to receiving your revised manuscript.

Kind regards,

Brett Austin Froelich, Ph.D

Academic Editor

PLOS ONE

Journal Requirements:

Reviewers' comments:

Reviewer's Responses to Questions

**Comments to the Author**

1. If the authors have adequately addressed your comments raised in a previous round of review and you feel that this manuscript is now acceptable for publication, you may indicate that here to bypass the “Comments to the Author” section, enter your conflict of interest statement in the “Confidential to Editor” section, and submit your "Accept" recommendation.

Reviewer #1: (No Response)

Reviewer #2: All comments have been addressed

2. Is the manuscript technically sound, and do the data support the conclusions?

Reviewer #1: Yes

Reviewer #2: (No Response)

3. Has the statistical analysis been performed appropriately and rigorously? 

Reviewer #1: N/A

Reviewer #2: (No Response)

4. Have the authors made all data underlying the findings in their manuscript fully available?

Reviewer #1: Yes

Reviewer #2: (No Response)

5. Is the manuscript presented in an intelligible fashion and written in standard English?

Reviewer #1: Yes

Reviewer #2: (No Response)

6. Review Comments to the Author

Reviewer #1: The authors of “A systematic review of waterborne and water-related disease in animal populations of Florida from 1999-2019” made several changes to improve the manuscript’s quality, clarity, and adherence to PLOS ONE’s guidelines for systematic reviews. The inclusion of the new map figure, table, and clarification of the terminology around toxins v. pathogens were especially appreciated. Specific comments and suggestions on the revised manuscript are noted below.

Abstract

• Line 44: Clarify that these are not necessarily the diseases that have been circulating in animals, but those that have been reported on.

Introduction

• Line 75: Include an example of a vector here; the author’s include examples of vectors below (line 80), but suggest moving this to line 75 where vectors are first mentioned.

• Paragraph beginning at line 78: Seems a bit repetitive with the preceding paragraph (definitions of toxins, pathogens, etc.). Suggest combining these two paragraphs to improve flow. It’s also not clear why toxins are mentioned in the preceding paragraph but not the one starting at line 78.

• Lines 86-88: Suggest also mentioning that some of these diseases are reportable because they’re there is a need to control for potential person-to-person transmission after zoonotic infection.

• Lines 94-99: Suggest rewording or adding a sentence that explicitly states that while you’re reporting on the number of studies, this does not necessarily correlate to risk or prevalence associated with a given disease agent.

Results

• The authors have made a number of revisions to clarify pathogens v. toxins in the manuscript. However, in lines 211 and 219 and in Table 2, K. brevis is seemingly referred to as a toxin. Suggest clarifying throughout that K. brevis is a toxin-producer and not a toxin or pathogen itself and differentiating toxins (like saxitoxin) from toxin-producers (like K. brevis). For example, the authors could change “frequently examined pathogens and toxins” to “frequently examined pathogens and toxin-producers” in line 211.

Discussion

• Suggest adding subheadings.

• Line 261: Suggest changing “found” to “reported”.

• Line 263: Use abbreviation E. coli here since the full taxonomic name is listed in line 261.

• Line 264: Suggest using more precise language to describe E. coli; E. coli is a species with subtypes (strains) that are pathogenic.

• Line 266: FDOH abbreviation not previously defined.

• Line 269: If pathotype wasn’t described, how did the study authors use E. coli? Perhaps as a fecal indicator? This wouldn’t necessarily fit in with your description of E. coli as a pathogen, but any detectable E. coli might be problematic since it is used as fecal indicator and feces in general carry a wide array of disease agents (Salmonella, Campy, norovirus, etc.). The authors seem to be hinting at this later in the same paragraph, but this should be clarified if E. coli is included as a specific discussion point.

• Line 271 to end of paragraph: Is this section intended to be specific for E. coli or all bacterial enteropathogens? Please clarify.

• Suggest adding additional detail to the discussion of the places where you found the most studies of various animals and diseases (line 435). The map shows some interesting geographic trends with regard to the types of animals and pathogens/toxins being looked at and the total number of studies from in various parts of the state. Is this distribution related ecology? Funding? University locations? Population size? What areas might benefit from additional research based on the results of this study?

Figures

• It may just be the pdf version created for reviewers, but both figures are poor resolution and difficult to read; suggest improving figure resolution.

Reviewer #2: (No Response)

7. PLOS authors have the option to publish the peer review history of their article (what does this mean?). If published, this will include your full peer review and any attached files.

Reviewer #1: No

Reviewer #2: **Yes: **A DENENE BLACKWOOD

---

## [Author Response · Author response to Decision Letter 1]

29 Jun 2021

Please see attached document titled, "PLOS ONE rev 2_June2021_Comments.docx" for an outline of each reviewer comment and author response.

---

## [Editor Report · Decision Letter 2]

9 Jul 2021

A Systematic Review of Waterborne and Water-Related Disease in Animal Populations of Florida from 1999-2019

PONE-D-20-32325R2

Dear Dr. Barnes,

We’re pleased to inform you that your manuscript has been judged scientifically suitable for publication and will be formally accepted for publication once it meets all outstanding technical requirements.

Kind regards,

Brett Austin Froelich, Ph.D

Academic Editor

PLOS ONE

---

## [Editor Report · Acceptance letter]

13 Jul 2021

PONE-D-20-32325R2 

A Systematic Review of Waterborne and Water-Related Disease in Animal Populations of Florida from 1999-2019 

Dear Dr. Barnes:

I'm pleased to inform you that your manuscript has been deemed suitable for publication in PLOS ONE. Congratulations! Your manuscript is now with our production department. 

Kind regards, 

on behalf of

Dr. Brett Austin Froelich 

Academic Editor

PLOS ONE